# A circular white-light flare with impulsive and gradual white-light kernels

Q. Hao [1,2], K. Yang [1,2], X. Cheng [1,2], Y. Guo [1,2], C. Fang[1,2], M.D. Ding[1,2], P.F. Chen [1,2] & Z. Li[1,2]

White-light flares are the flares with emissions visible in the optical continuum. They are thought to be rare and pose the most stringent requirements in energy transport and heating in the lower atmosphere. Here we present a nearly circular white-light flare on 10 March 2015 that was well observed by the Optical and Near-infrared Solar Eruption Tracer and Solar Dynamics Observatory. In this flare, there appear simultaneously both impulsive and gradual white-light kernels. The generally accepted thick-target model would be responsible for the impulsive kernels but not sufficient to interpret the gradual kernels. Some other mechanisms including soft X-ray backwarming or downward-propagating Alfvén waves, acting jointly with electron beam bombardment, provide a possible interpretation. However, the origin of this kind of white-light kernel is still an open question that induces more observations and researches in the future to decipher it.

[1] School of Astronomy and Space Science, Nanjing University, Nanjing 210023, China. [2] Key Laboratory of Modern Astronomy and Astrophysics (Nanjing University), Ministry of Education, Nanjing 210093, China. Correspondence and requests for materials should be addressed to Q.H. (email: haoqi@nju.edu.cn) or to C.F. (email: fangc@nju.edu.cn) or to M.D.D. (email: dmd@nju.edu.cn)

Solar flares result from a sudden release of magnetic energy in the solar atmosphere[1,2]. In very few cases, solar flares can also show an enhanced emission at the visible continuum, besides the enhanced spectral lines that are formed in the chromosphere and above. These flares are called white-light flares[3]. The flare observed for the first time in 1859 was actually a white-light flare[4]. However, the number of reported white-light flares only occupies a small fraction of the total number of solar flares observed so far. Traditionally, enhanced white-light emission is detected in the Balmer and Paschen continuum. But in a few cases, enhanced emission is also observed in the infrared continuum[5–8]. White-light flares are classified into two different types, types I and II[9]. The different characteristics of each type were studied by using both observations and atmospheric modeling[10]. For type I white-light flares, there is a good time correlation between the maximum of the continuum emission and the peaks of the hard X-ray (HXR) and microwave emissions, and there exists a strong Balmer jump in the spectra. Besides, the Balmer lines are usually strong and broad. However, type II white-light flares appear less frequently and do not display the above features[11–15].

It is generally believed that the continuum emission of white-light flares comes mainly from the lower chromosphere down to the middle photosphere. On the other hand, the energy release is often considered to take place in the corona. Therefore, how energy is transported to the lower atmosphere to produce the white-light emission has always been an open question[16,17]. Interestingly, many white-light flares have been observed on dKe/dMe stars by the Kepler mission[18]. The stellar flares can be $10–10^7$ times more energetic than the solar white-light flares[19,20], since there is a big diversity in the magnetic field and flaring area on the stellar surface[21]. In spite of this diversity in physical parameters, study of solar white-light flares can provide a reference for study of stellar flares.

The elongated structures of solar flare emissions in the chromosphere are called flare ribbons. The morphology and evolution of flare ribbons have been interpreted in terms of the classical two-dimensional flare model called the CSHKP model[22–25]. Nevertheless, many flares take place in a more complicated three-dimensional (3D) structure, such as magnetic configurations containing 3D null points. Magnetic field lines associated with a 3D null point usually display a fan-spine configuration[26,27]. The dome-shaped fan portrays the closed separatrix surface, and the inner and outer spine field lines in different connectivity domains meet at the null point. When reconnection takes place at the null point or the separatrix layer, flare emissions at the footpoints of the fan field lines would constitute a closed or open circle-like flare ribbon. Since the first comprehensive study of a circular ribbon flare was carried out[28], evidence of circular ribbon flares has been reported by several authors[29–33]. However, up to now no event reported in the literature shows circular ribbons in white-light.

In this paper, we report a very rare circular white-light flare that consists of several bright kernels in the white-light continuum, as well as in the Hα line and ultraviolet (UV) channels. In particular, this flare possesses two categories of white-light kernels, i.e., impulsive kernels and gradual kernels. The two categories of kernels show different features in time evolution and spectral distribution of the white-light continuum, as well as different correspondences to HXR emissions. The basic magnetic structure of the flare region is a nearly enclosed dome with some flux ropes imbedded below. The circular flare ribbon, distinct white-light kernels, and peculiar magnetic structure make this flare a unique event among the white-light flares observed so far.

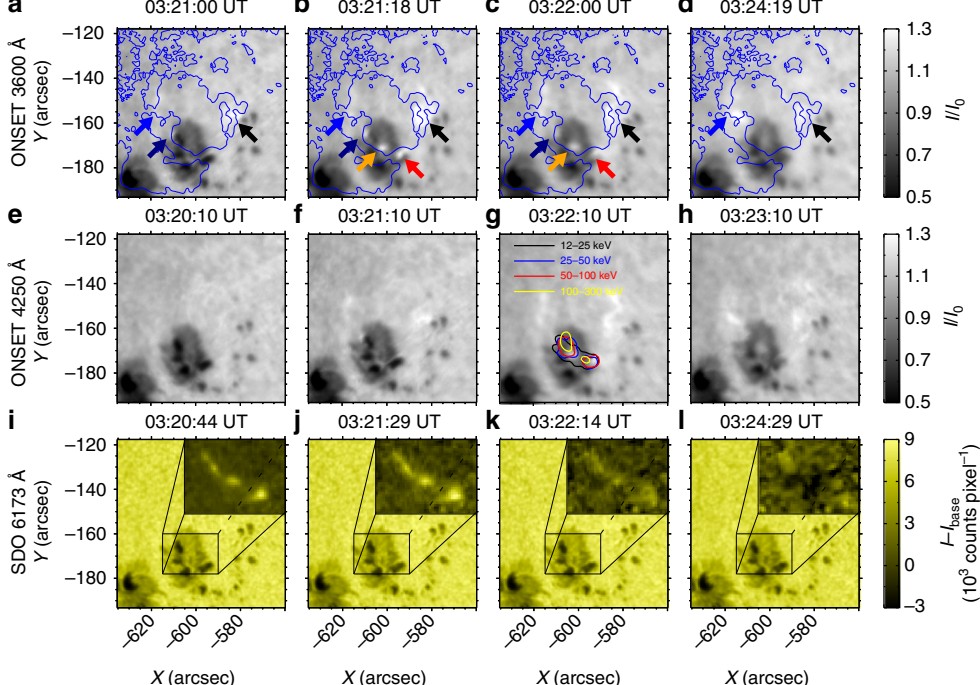

**Fig. 1** Images of the white-light flare around the flare peak time. **a–d** ONSET 3600 Å images. **e–h** ONSET 4250 Å images. **i–l** SDO 6173 Å images. The field of view of each panel is 75″ × 75″. The arrows in **a–d** indicate the locations where the white-light emission occurred. Blue lines in **a–d** mark the polarity inversion line of the line-of-sight magnetogram. The RHESSI HXR contours of 40% the maximum intensity in the energy bands 12–25, 25–50, and 50–100 keV are overplotted in **g** at roughly the flare peak time. The contour is 60% for the energy band 100–300 keV. The quantity $I_0$ in **a–h** refers to the background intensity in a nearby quiet region. The enlarged frames in **i–l** are the base-difference images of 6173 Å, relative to the base image at 03:19:59 UT. The counts in **i–l** are measured for each exposure

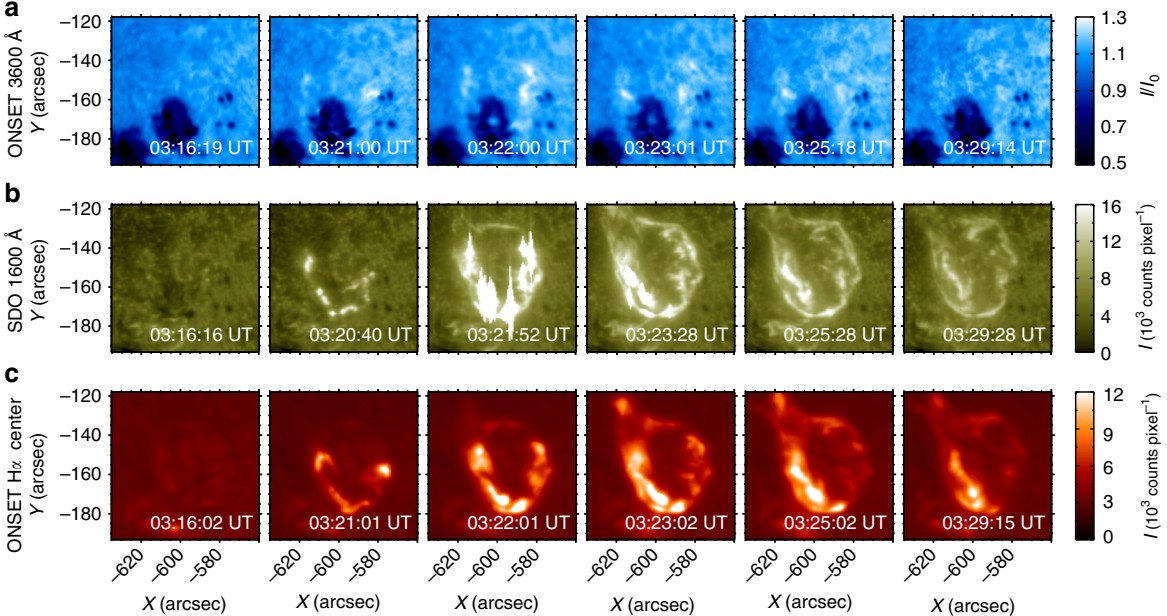

**Fig. 2** Images showing the flare morphology and evolution. **a** Images at 3600 Å by ONSET. **b** Images at 1600 Å by SDO/AIA. **c** Images at Hα line center by ONSET. The field of view of each image is the same as that of Fig. 1. These images show how the bright kernels and the circular ribbon form and evolve and a comparison of the different emissions at 3600 Å, 1600 Å and Hα line center. The quantity $I_0$ in **a** refers to the background intensity in a nearby quiet region. The counts in **b**, **c** are measured for each exposure

## Results

**Overview of the flare**. A white-light flare with a GOES class M5.1 occurred at S15E39 in active region 12297 on 10 March 2015. The flare was well observed by the Optical and Near-infrared Solar Eruption Tracer (ONSET)[34]. The flare started at about 03:19 UT and ended near 03:30 UT. We analyze the ONSET data observed from 03:13 to 03:35 UT at 3600 Å, 4250 Å, and Hα line center (see Methods section). The pixel size is about 0.24″ and the cadence is 15–60 s depending on the observation modes. Some frames are neglected when the seeing condition becomes bad temporarily. We also analyze the UV emission from the Atmospheric Imaging Assembly (AIA)[35] and the magnetic field data from the Helioseismic and Magnetic Imager (HMI)[36,37] on board Solar Dynamics Observatory (SDO)[38]. Figure 1 shows the flare evolution at 3600, 4250, and 6173 Å around the peak time of the flare. One can see from the 3600 Å images that there are three bright kernels at 03:21 UT, as pointed out by the blue, navy, and black arrows (Fig. 1a). Eighteen seconds later, two more bright kernels, as indicated by the orange and red arrows, also brightened (Fig. 1b, c). These bright kernels form a semicircular morphology. The kernel indicated by the blue arrow brightened for a relatively long time and was much brighter than the others at 03:24 UT (Fig. 1d). Figure 1e–h shows a similar process happening at 4250 Å. We also find a white-light enhancement at SDO/HMI 6173 Å (Fig. 1i–l). Since the white-light enhancement at 6173 Å is relatively weak, we also present the base-difference image of 6173 Å around the peak time of the white-light flare, from which the white-light kernels can be seen more clearly. Unlike the features revealed by the two wavebands of ONSET, there are several white-light kernels located in the penumbral regions in the SDO/HMI 6173 Å continuum image. We also reconstruct the HXR images from the data observed by Reuven Ramaty High-Energy Solar Spectroscopic Imager (RHESSI)[39] (see Methods section). Taking into account the uncertainties caused by the spatial resolution and the projection effect, the white-light kernels marked by the orange and red arrows match well with the centroid of HXR contours. In particular, we find a close spatial correspondence between the continuum emission and the HXR emission in the energy band of 100–300 keV at 03:22 UT (yellow contour in Fig. 1g). However, there are no obvious HXR sources associated with the white-light kernels marked by the blue and black arrows.

Figure 2 shows the morphology and evolution of the circular ribbon flare at 3600 Å, 1600 Å, and Hα line center. It is seen from the 1600 Å images that, about 5 min prior to the white-light flare, some small brightenings appeared at 03:16 UT in the southeastern part of the field of view (Fig. 2b). Then some particular kernels within the circular flare ribbon started to brighten at around 03:21 UT (Fig. 2). A few minutes later, the brightness of the two southern kernels increased drastically nearly simultaneously, which then developed as bright threads. The upper one formed an inner ribbon and the lower one expanded and formed an outer circular ribbon at 03:23 UT (Fig. 2b, c). These bright kernels in 1600 Å and Hα correspond well to the locations of the enhanced white-light emission. In particular, the eastern bright kernel, which is marked by the blue arrow, started brightening earlier and lasted longer than the other kernels (Fig. 1a–d). Even at a late time of 03:25 UT, this bright kernel was still visible (Fig. 2a). In order to show the overall evolution of the flare, we also display some snapshots in the preflare and postflare phases in Fig. 2.

**Light curves of the flare**. Figure 3 shows the light curves of the flare at different wavebands. It can be seen that the impulsive phase commences simultaneously at around 03:21 UT at different energy bands of HXR. The peak time of GOES soft X-ray (SXR) flux (1–8 Å) is about 03:24 UT, 3 min later than the HXR peak time. In particular, we find that the peak of the time derivative of the SXR flux is roughly coincident with the HXR peak, implying that the Neupert effect applies well to this event[40,41]. As mentioned above, there are five white-light enhancement kernels. In order to analyze the detailed evolution of the white-light emission at these kernels, we calculate the contrast $C = (I − I_0)/I_0$

at H$\alpha$, 4250, and 3600 Å, where $I_0$ is the background intensity in the quiet region near the flare site. However, the preflare contrast is not zero at some sites like the two bright kernels in the penumbral region as indicated by the orange and red arrows (Fig. 1b, c). Therefore, the preflare value, adopted as the mean value during 03:13 to 03:19 UT, is then subtracted from the contrast at each kernel. The light curves in H$\alpha$, 4250 Å, and 3600 Å for each kernel are shown in Fig. 3b–d. It is seen that the light curves at the two continuum wavebands have a similar evolution trend. Generally speaking, the white-light emissions at the

positions marked by the blue and navy arrows evolve more gradually, whereas those marked by the orange and red arrows rise more impulsively (note that the light curves and the arrows have a one-to-one correspondence in color).

In order to make a quantitative comparison, we measure the lifetime and the rise time of the white-light emissions at different kernels. The lifetime is defined as the time interval from $1/e$ of the peak contrast in the rising phase to that in the decay phase. The rise time is defined as the time interval in the rising phase from $1/e$ to the peak contrast. The results are summarized in Table 1, which shows that different white-light kernels have quantitatively different time evolutions. According to the different lifetimes and rise times, we divide the white-light kernels into three different categories, including impulsive, gradual, and intermediate kernels. In addition, we also give the onset time of each feature in Table 1 for reference. The onset time is defined as the first time when the light curve shows significant monotonic increase. Owing to the low cadence of white-light observations, we can only give a possible time range as the onset time of the white-light emissions.

**Different categories of white-light kernels.** The rise time of the two kernels marked by the orange and red arrows is <20 s, similar to that of the HXR flux (Table 1). These two impulsive kernels exhibit a dramatic increase in the continuum contrast, whose peak correlates well with the HXR peak, in particular at the energy band of 100–300 keV. Moreover, the impulsive kernels match well with the centroid of HXR contours (Fig. 1g). These suggest a close relationship between the energetic electrons and the white-light emission. Furthermore, the two impulsive kernels have opposite magnetic polarities but a similar time evolution in continuum enhancement.

The lifetimes and rise times of the gradual kernels are much longer than that of the impulsive kernels and that of the HXR flux (Table 1). Besides, the continuum emission at the gradual kernels is relatively weak and without any obvious HXR sources that are spatially correlated with them (Fig. 1g). Between the two gradual kernels, there still exist some differences in the sense that the maximum white-light continuum of the kernel shown by the blue arrow lags behind the HXR peak (nearly correlates with the SXR peak), whereas the other kernel, indicated by the navy arrow, presents a white-light emission peak nearly coincident with the HXR peak (Fig. 3a, d). However, the lifetime of the kernel indicated by the navy arrow is much longer than that of the HXR emission. Therefore, despite a relatively quick rise of the white-light emission at this kernel, its long lifetime implies that the origin of this kernel is basically similar to the kernel marked by the blue arrow. In principal, the similarity between the SXR and white-light emissions at the gradual kernels indicates a mainly thermal contribution to the heating of these kernels[42].

By comparison, the kernel marked by the black arrow shows a white-light time profile with a nearly impulsive rising phase but a relatively long decay phase (Fig. 3d). The maximum continuum

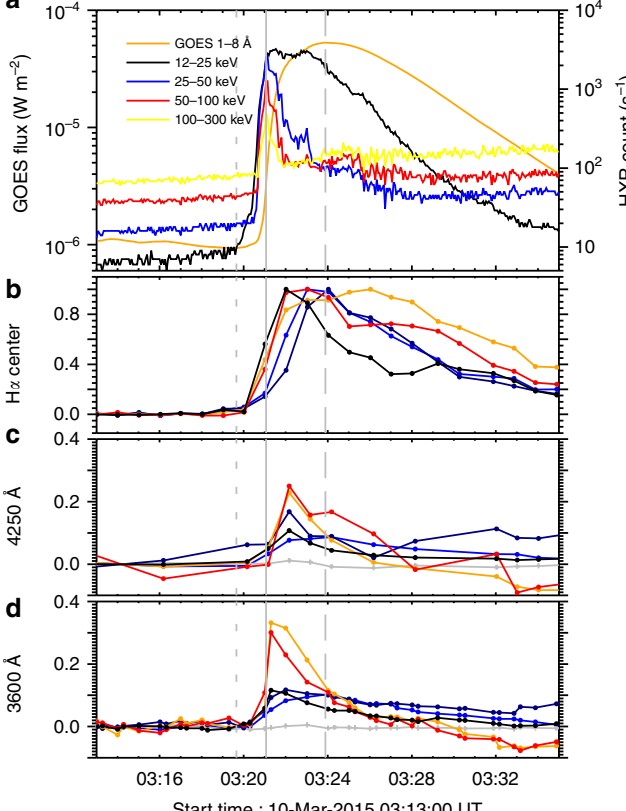

**Fig. 3** Light curves of the flare at different wavebands. **a** The GOES SXR 1–8 Å flux (orange) and the RHESSI HXR counts at 12–25 keV (black), 25–50 keV (blue), 50–100 keV (red), and 100–300 keV (yellow) for the white-light flare. **b–d** Relative enhancements (contrasts) at H$\alpha$ line center, 4250, and 3600 Å for the areas where the white-light emission appears. The colors of the curves correspond to the colors of the arrows in Fig. 1b. The short dashed vertical lines show the onset time of the SXR flux, the solid vertical lines show the peak time of the HXR flux, and the long dashed vertical lines show the peak time of the SXR flux. The gray lines with plus signs in **c**, **d** show the continuum contrast of a plage near the flare site at 4250 and 3600 Å, respectively

**Table 1 Characteristic times of the X-ray fluxes and 3600 Å white-light emission at five flare kernels**

|  | HXR | | SXR | Impulsive kernels | | Gradual kernels | | Intermediate kernel |
|---|---|---|---|---|---|---|---|---|
|  | 25–50 keV | 50–100 keV | 1–8 Å | Red | Orange | Blue | Navy | Black |
| Onset time[a] (s) | 49 | 57 | 0 | ~49–81 | ~49–81 | ~49–81 | ~49–81 | ~49–81 |
| Rise time[b] (s) | 16.1 | 15.4 | 138.8 | 17.8 | 13.7 | 179.8 | 61.3 | 33.8 |
| Lifetime[c] (s) | 45.5 | 41.0 | 465.2 | 178.3 | 171.8 | 550.3 | 631.9 | 297.5 |

[a] Defined as the elapsed time from the onset time of the SXR emission at 03:19:39 UT
[b] Defined as the time interval from $1/e$ to the peak contrast
[c] Defined as the time interval from $1/e$ of the peak contrast in the rising phase to that in the decay phase

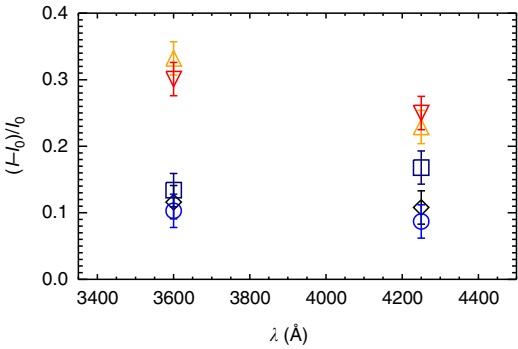

**Fig. 4** Maximum continuum contrasts at the white-light kernels. The data are measured for the five kernels and the two wavebands (3600 and 4250 Å). The colors of the symbols correspond to the colors of the arrows in Fig. 1b. The vertical bars represent an observation error of 2.5% at both wavebands (see Methods section for details about the error estimation)

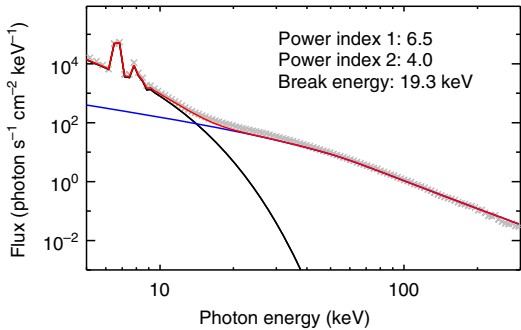

**Fig. 5** HXR spectrum of the flare and its fitting. The integration time of the spectrum is from 03:20:56 to 03:21:20 UT. The spectrum is fitted using a variable thermal source plus a non-thermal source with a broken power law. The gray cross signs refer to the spectral data after subtraction of the background. The modeled thermal, non-thermal components, and the overall spectrum are plotted in black, blue, and red colors, respectively

emission appears at around 03:21 to 03:22 UT, which coincides roughly with the peak HXR emission. However, we still do not find any obvious HXR sources that are spatially correlated with this kernel (Fig. 1g). Considering its short rise phase but long decay phase, we regard this kernel as an intermediate type with probably a combination of both impulsive and gradual origins.

**Balmer jump for the impulsive kernels.** Since the two continuum wavebands 3600 and 4250 Å are within and outside the Balmer continuum, respectively, we are able to figure out whether these white-light emissions possess a Balmer jump or not. For this purpose, we check the maximum continuum contrasts at the two continuum wavebands. The results are plotted in Fig. 4. We find that the maximum enhancement at 3600 Å is significantly larger than that at 4250 Å for the impulsive white-light kernels. We need to check whether such different continuum enhancements at the two wavelengths can be fitted by a blackbody emission. A quantitative parameter for judgment is the contrast ratio, $C_{3600}/C_{4250}$. From observations, this ratio amounts to 1.45 and 1.20 for the two kernels indicated by the orange and red arrows, respectively. However, the blackbody emission from an emitting source with a reasonable temperature range of, say, 5000–10,000 K, predicts a contrast ratio no larger than 1.18. This implies that the white-light emissions at these kernels can unlikely be accounted for by an isothermal blackbody source and that there exists a Balmer jump in the continuum spectra. By contrast, one can see that, for the gradual kernels, the maximum enhancement is nearly the same at 3600 and 4250 Å (the blue circles) or the enhancement at 4250 Å is even slightly larger than that at 3600 Å (the navy squares). Obviously, no Balmer jump appears at these gradual kernels, contrary to the case of impulsive kernels.

**Energy spectrum of the HXR emission.** We further derive the energy spectrum of the HXR emission. The spectrum is integrated from 03:20:56 to 03:21:20 UT. We fit the spectrum from 3 to 300 keV using a broken power law with a thick-target component plus a thermal component. The fitted results are shown in Fig. 5. The break energy is 19.3 keV. The spectral indices of the electron beam below and above the break energy are 6.5 and 4.0, respectively. We can see that the electron flux at the high energy band is much higher than in ordinary flares. For the impulsive white-light emission revealed in the current flare, the spectral indices of the HXR emission are in the range of the statistical results[43].

A statistical analysis of 43 flares spanning GOES classes M and X indicates that the X-ray flux at 30 keV or the electron flux at 50 keV is best correlated with the white-light flux at 6173 Å[44]. We calculate the white-light flux in our event using a similar method. We choose the SDO/HMI 6173 Å preflare image at 03:19:59 UT and the peak-time image at 03:21:29 UT for analysis. The difference image is shown in Fig. 1j, from which the white-light flux is calculated to be about $6 \times 10^5$ DN s$^{-1}$. The HXR photon flux at 30 keV is 25.7 photon s$^{-1}$ cm$^{-2}$ keV$^{-1}$. We find that quantitatively the parameters of our event lie within the range of the statistical results[44].

**Magnetic field configuration of the flare.** Using the vector magnetic field on the photosphere from SDO/HMI, we make a nonlinear force-free field extrapolation applying the optimization method[45] to obtain the magnetic field configuration (see Methods section). The 3D magnetic field is much more complex than a typical symmetric null point configuration. There is neither a null point nor an inner spine since the polarity inversion line possesses an irregular path, especially in the northeastern part (Fig. 6a). However, owing to the nearly enclosed polarity inversion line, there still exists a dome structure, which is similar to the previous observations[46]. To study the magnetic topology for the circular ribbon flare, we employ the concept of quasi-separatrix layer (QSL), which is defined as a layer with dramatic magnetic connectivity variation (see Methods section). The QSL is usually quantified by the squashing factor $Q$. It is seen that the QSL, where $Q$ has large values, is almost cospatial with a dome structure (Fig. 6b). This phenomenon has been observed in many previous studies[29,30,33,47]. We draw some selected field lines portraying the dome structure, which are located in the large $Q$ value regions (Fig. 6b). The flare appears under the large dome.

## Discussion

From the extrapolated magnetic field, we find that the impulsive white-light kernels are a pair of conjugate footpoints, located under a twisted flux rope with relatively strong magnetic field strength (red lines in Fig. 6). Magnetic reconnection may take place between the field lines that are highly twisted within the flux rope, so that energetic electrons could propagate along the field lines and bombard the flux rope footpoints. The similar situations were already observed and simulated before[48,49].

The features of the two impulsive kernels consolidate the proposition that the energetic electrons with relatively high energies produce the impulsive white-light emission. In the classical thick-target model, electrons of energies, say, about 50 keV, cannot penetrate to the photosphere or the lower chromosphere, and thus backwarming may be a supplementary, yet

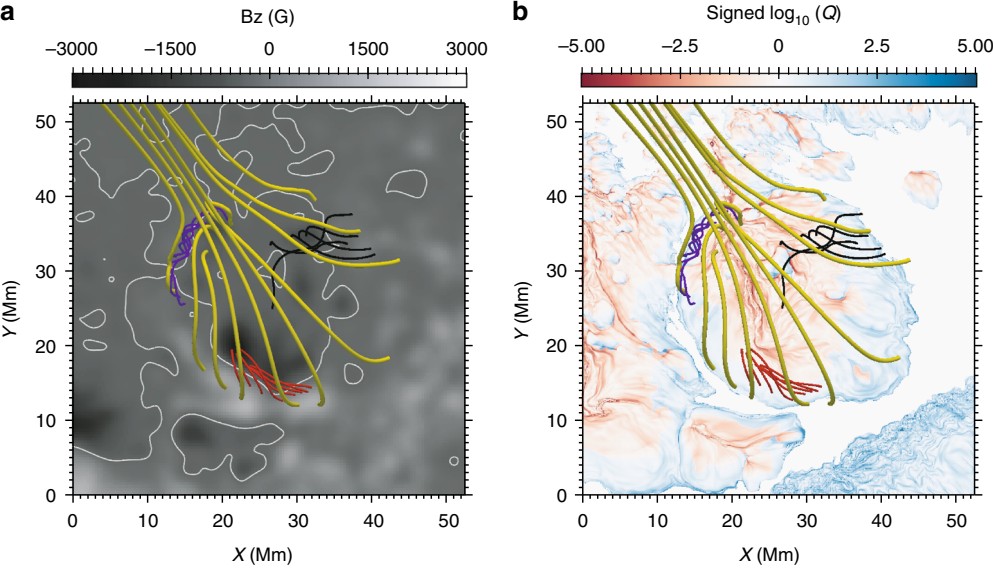

**Fig. 6** Magnetic field configuration of the flare region. **a** Map of the photospheric vertical magnetic field. White contours mark the polarity inversion line of the magnetogram. **b** Map of the squashing factor $Q$ on the bottom boundary calculated from the nonlinear force-free field. The sign of $\log_{10} Q$ is taken from that of the corresponding magnetic polarity. Small flux ropes colored with blue, red, and black appear under the large dome structure colored with yellow in **a**, **b**. The field of view is equivalent to that of Fig. 1

necessary, mechanism for the heating in the lower atmosphere and the generation of the white-light emission. However, we need to point out that, in our event, the electron beam flux is still much higher at energies over 100 keV as compared with other ordinary flares. This implies that there is a possibility that electrons with very high energies can directly penetrate to the lower chromosphere and even below. If this is the case, the electrons can directly produce heating and induce white-light emission there. Although we cannot quantitatively judge if the energy residing in those high energy electrons is sufficient, we postulate that direct heating plays at least a partial role in addition to the backwarming effect in the lower atmosphere.

The continuum emission rises more gently and lasts longer in the gradual white-light kernels, though with a lower contrast (blue and navy light curves in Fig. 3c, d). These white-light kernels show up as ribbons and patches in relatively large areas (Fig. 1). It indicates that the heating is less intensive but continues after the flare peak in these regions. In fact, gradual white-light kernels have been reported before[42]. Those reported white-light kernels are found to be correlated in space with HXR sources, though there is a time lag of tens of seconds of the peak white-light emission with respect to the peak HXR emission[42]. Nevertheless, in our event, we did not find any HXR sources that are correlated to the gradual white-light kernels, implying that electron beam bombardment is at least not the main cause for them. In addition, no significant Balmer jump is present at these kernels as revealed by Fig. 4. It means that these kernels could be due to the negative hydrogen (H⁻) emission as in the case of type II white-light flares. The H⁻ emission originates in relatively deep layers like the upper photosphere[13–15,50]. Since electron beam bombardment is unlikely the main heating source, heating of such lower layers becomes more challenging than in the case of type I white-light flares.

Among the gradual kernels, the most distinguished one is plotted with the blue curve. It shows both a gentle rising phase and a gentle decay phase with a quite long lifetime. The maximum continuum emission appears even 3 min later than the peak HXR emission. In two previous studies, it was found that the X-ray and γ-ray emissions peak about 1 min before the white-

light emission[12], or the continuum emission can precede the HXR emission and the radio burst[15]. However, these two white-light flares, which were identified as type II white-light flares, are impulsive events with short durations. In our event, while the gradual white-light kernel shows some features of type II white-light flares, such as no Balmer jump, it does present new features, i.e., the slow rise and long lifetime of the white-light emission, which were not found in most previously reported type II white-light flares. Such features imply a specific scenario of energy release, as discussed below.

The extrapolated magnetic field shows that the gradual and intermediate kernels are located at the footpoints of peripheral twisted field lines with relatively weak magnetic field strength (blue and black lines in Fig. 6), which are far away from the core twisted field lines with strong magnetic field strength (red lines in Fig. 6). The QSL reconnection may happen around the flux ropes[49], not only in the dome structure as in the bright point case[51]. In our case, QSL reconnection may take place in several peripheral twisted field lines producing separate white-light emissions at the gradual kernels and also possibly the intermediate kernel in the beginning (the three kernels pointed by the blue, navy, and black arrows in Fig. 1a). These reconnections together with the reconnection in the dome probably facilitate the eruption of the core twisted field lines, which leads to two impulsive white-light kernels at the conjugate footpoints of a flux rope (the two kernels pointed by the orange and red arrows in Fig. 1b, c). The eruption then disturbs the dome structure above and the lower layer magnetic field configuration, further sustaining the QSL reconnection for a long time to generate the continuous gradual white-light emissions. It is possible that, in the early phase of the QSL reconnection, some energetic electrons are also generated in the lower atmosphere, which are thermalized locally during a very short time because of the high density there. The local thermalization results in a heating that may explain the relatively rapid increase of the white-light emission at the intermediate kernel (the kernel pointed by the black arrow in Fig. 1b, c). Note that the flux of such energetic electrons may be quite low so that they cannot yield an observable HXR source.

Although the QSL reconnection may occur at a site somewhat lower than the corona as usually thought, where the exact reconnection site is and how the released energy is transported to the white-light emission layer are difficult to be determined with the present data. They remain open questions. We conjecture two possible mechanisms for the origin of the gradual kernels. If excluding the electron beam as the main energy carrier, SXR backwarming is a potential candidate, considering the similarity between the SXR and white-light time evolutions[42]. Another possibility is heating by Alfvén waves[52]. The QSL reconnection around the flux tube may excite fairly strong Alfvén waves, which propagate to the lower atmosphere and deposit energy there. Unfortunately, the present observations are not sufficient in testifying existence of Alfvén waves. We expect future coordinated observations of white-light flares, in particular including high-cadence spectral observations of chromospheric lines like Hα, whose periodic shifts may imply the existence of Alfvén waves[53]. Besides the observations, radiative hydrodynamic simulations are also required to elucidate the different roles of energy transport mechanisms, including electron beam bombardment, heat conduction, Alfvén waves, and the radiative backwarming. In fact, it is quite possible that two or more mechanisms work together in a single event.

In summary, we report a circular white-light flare on 10 March 2015 that shows some peculiar features. Through analyzing the white-light emissions at 3600, 4250, and 6173 Å wavebands, as well as the 3D magnetic configuration, we find two different kinds of white-light emission kernels, one being impulsive with a close relationship to the HXR emission, and the other being gradual without spatially correlated HXR sources. In particular, the origin of the gradual kernels requires mechanisms other than the electron beam bombardment. The new features of this event pose a challenge to the white-light flare models and await more researches both in observations and numerical simulations.

## Methods

**Instrumentation**. The ONSET telescope is installed near the Fuxian lake near Kunming, China, jointly operated by Nanjing University and Yunnan Astronomical Observatory. It consists of four tubes, which allow the quasi-simultaneous observations of the Sun in four wavebands, He I 10830 Å, Hα, and 3600 and 4250 Å continua. The Hα vacuum tube has an aperture of 27.5 cm and works at the Hα line center and wings up to ±1.5 Å with a band width of 0.25 Å. The white-light vacuum tube has an aperture of 20 cm, working at the 3600 and 4250 Å continua with a band width of 15 Å. The cadence is 15–60 s depending on the observation modes. Up to now, ONSET has observed a number of white-light flares since its operation[54,55].

**Error estimation of white-light observation**. We select a quiet region with a size of $150 \times 150$ pixels near the flare site and then calculate the mean standard deviation of the intensity from 03:12 to 03:42 UT in order to estimate the observation error in the continua. The mean standard deviations are $2.44 \times 10^3$ and $7.60 \times 10^3$ counts s$^{-1}$ pixel$^{-1}$ for 3600 and 4250 Å, respectively. They correspond to an observation error of about 2.5%.

**HXR image reconstruction**. We employ the Pixon algorithm[56] to reconstruct the HXR sources from the data observed by RHESSI. The detectors 1, 3, 5, 6, 7, and 9 are used in the image reconstruction. The spatial resolution is about 3″ and the integral time is 1 min. We reconstruct the HXR sources in different energy bands of 12–25, 25–50, 50–100, and 100–300 keV in the impulsive phase. The HXR contours at different energy bands are overplotted in Fig. 1g.

**Nonlinear force-free field extrapolation**. In order to study the magnetic structure of the flare region, we make a nonlinear force-free field extrapolation for a region around the flare site at a time of 03:12 UT before the occurrence of the white-light flare. The boundary data are the vector magnetic field on the photosphere observed by SDO/HMI. We employ the optimization method[45]. The method is based on the idea of minimizing the functional that combines the magnetic field divergence, the Lorentz force, and the error in the observations. The functional is defined as,

$$\mathcal{L} = \int_\Omega \left( \omega_d |\nabla \cdot \mathbf{B}|^2 + \omega_f \frac{|(\nabla \times \mathbf{B}) \times \mathbf{B}|^2}{B^2} \right) dv + \nu \int_{\partial\Omega} (\mathbf{B} - \mathbf{B}_{obs}) \cdot \mathbf{W} \cdot (\mathbf{B} - \mathbf{B}_{obs}) ds,$$

where d$v$ and d$s$ represent the differential volume and surface elements, respectively, and $W$ is a space-dependent diagonal matrix related to the measurement error. The weighting parameters $\omega_f$ and $\omega_d$ are chosen as unity in the region of interest and drop to zero inside the boundary layers, which contain 32 pixels. The line-of-sight component of $W$ is fixed to be unity and the transverse components are chosen to be the ratio between the transverse magnetic component and the maximum of transverse magnetic field. The parameter $\nu$ is chosen as 0.0001, which has been tested to be able to give the best numerical result[45]. It turns out that, in the region of interest, the average angle between the electric current and the magnetic field is 5.1°, and the measure of divergence of the magnetic field is $f_i = \int_{\Delta S_i} \mathbf{B} \cdot d\mathbf{S} / \int_{\Delta S_i} |\mathbf{B}| dS = 2.07 \times 10^{-5}$. It means that the result is acceptable.

**Calculation of the QSL**. To study the magnetic topology for the circular ribbon flare, we employ the concept of QSL, which is defined as a layer with dramatic magnetic connectivity variation. The QSL is usually quantified by the squashing factor Q, which is defined as the squared norm of the Jacobi Matrix of the field line mapping over the magnetic field strength ratio between the two footpoints of the field line, $Q = \frac{|\partial X / \partial x|^2}{B_0/B_1}$[57]. We calculate the Q distribution on the solar surface and plot the signed $\log_{10} Q$ distribution with the color scale in Fig. 6b, where the sign of $\log_{10} Q$ is taken from that of the corresponding magnetic polarity.

**Data availability**. All the data used in the present study are publicly available. The ONSET 3600 and 4250 Å continua and Hα images can be downloaded from http://sdac.nju.edu.cn/. The SDO/AIA UV data, SDO/HMI 6173 Å data and vector magnetograms can be downloaded from http://jsoc.stanford.edu. The GOES X-ray flux data can be downloaded from http://www.ngdc.noaa.gov/stp/satellite/goes/dataaccess.html. The RHESSI X-ray data can be downloaded from https://hesperia.gsfc.nasa.gov/hessidata/.

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

## Acknowledgements

We are grateful to the ONSET, RHESSI, and SDO teams for providing the observational data. We also thank Dr Wei Liu and Dr. Richard A. Schwartz for valuable discussions about the accuracy and spatial resolution of the reconstructed HXR images. This work was supported by NKBRSF under grants 2014CB744203; NSFC grants 11533005, 11703012, 11733003, 11722325, 11773016, and 11373023; and Jiangsu NSF grants BK20170629 and BK20170011. P.F.C. was also supported by Jiangsu 333 Project.

## Author contributions

Q.H. analyzed the observational data and wrote the first draft. C.F. and M.D.D. initiated the study, supervised the project, and led the discussions. K.Y. performed the magnetic field extrapolation. X.C., Y.G., P.F.C. and Z.L. joined the discussions. All authors contributed to revisions of the manuscript.
