## [Peer Review File · Nature Communications]

Reviewers' comments:

Reviewer #1 (Remarks to the Author):

Solar flares release tremendous amount of energy and are often associated with other important solar events, such as sunquakes, filament eruptions, and coronal mass ejections (CMEs). The white-light (WL) flares represent the most energetic component of flare emission. Their properties in lower solar atmosphere are not fully understood in terms of the intriguing particle acceleration and radiative transfer processes. This paper presents a WL flare observation with a circular ribbon, which is very interesting and was not reported before. Its research will extend the frontier of existing knowledge on the energy release process of flares and its impact on the deep solar atmosphere. But the paper was prepared a bit roughly and needs polishing before publishing in Nature Communications.

(1) Hinode had a good coverage of this event, in high resolution, with a good pre-flare vector data at 00 UT. Hinode observations may miss some part of the ribbons, but should be compared, especially in G-band.

(2) I am not convinced for the lack of HXR for the second WL peak as shown in Figure 3. The RHESSI HXR 12-25 keV plot has a second peak, and the 25-100 keV seems to be somehow strange. Is the attenuation of RHESSI considered? It is hard to believe that WL emission without HXR signature since HXR is very sensitive, even to the micro-flares.

(3) The authors identified the WL ribbon in the wavelengths of 360 nm and 425 nm. It is unable for me to evaluate the real contamination at these two specific observing channels due to lack of information on bandpass of the used filters.

(4) It is better to add a figure of Hinode/SP magnetogram marking the kernels.

(5) With the derived contrast, some quantitative analysis could be done, such as temperature.

(6) In order to form the dome-spine structure, the circular ribbon should enclose a compact opposite magnetic polarity core emission. The paper did not discuss this part at all.

(7) The authors state that the cadence of ONSET data is one minute, how can they plot sub-minute data points in the light curves of Figure 3?

Reviewer #2 (Remarks to the Author):

The main claim of this paper is "This is the first report of this type of gradual white-light emission, which is not correlated with the HXR emission." However, there is no discussion of the so-called Type II white-light flares that are not associated with HXR emission and appear in the late phase (e.g., Ding, M. D., Fang, C., & Yun, H. S. 1999, ApJ, 512, 454). Therefore, I cannot recommend publication in Nature Communications.

Furthermore, a statistical analysis of the 4250 and 3600 enhancements is completely absent from the paper, and it is not convincing from Figure 3 that the black, blue, and yellow light curves in 4250 experience a statistically significant gradual increase (e.g., how do the light curves of other regions of plage look compared to these light curves?). In the black, blue, and yellow curves in the 3600 light curve, only the black appears to increase, and this increase occurs at the same time as the red and orange increase (thus all would be impulsive). There are also many statements about the sources of energy deposition (in general and specifically to this flare) that are qualitative, without appropriate references, and unjustified.

Reviewer #3 (Remarks to the Author):

Referee's Report

NCOMMS-16-29810: A Circular White-Light Flare with Impulsive and Gradual White-Light Sources, by Hao, Q., Yang, K., Cheng, X., Guo, Y., Fang, C., Ding, M. D., Chen, P. F. and Li, Z.

The manuscript reports on an apparently rare solar flare that shows an observable signature in white light. While white-light flares statistically belong to the largest flares (GOES M- and X-classes), and hence represent a slim minority of the solar flare population, the rarity of the flare studied here is a roughly circular white-light ribbon consisting of two parts: an impulsive one, correlating with the flare's X-ray emission, and a gradual one, with no apparent correlation to X-rays. The gradual white-light flare emission, contributing much of the circular ribbon, is interpreted as either (slow) magnetic reconnection or downward-propagating Alfvén waves originating in the corona, with both mechanisms leading to emission along the circular footprint of a magnetic quasi-separatrix layer (QSL) bounding the entire flaring magnetic configuration.

The manuscript's topic is quite interesting, while analysis combines science-grade data from an array of ground- and space-based instruments: Near-Infrared Solar Eruption Tracer (ONSET), RHESSI and two telescopes (HMI and AIA) onboard the Solar Dynamics Observatory (SDO) mission. At the same time, it uses a nonlinear force-free (NLFF) field extrapolation for the analysis of QSLs. Its originality could potentially make it worthy of publication at Nature Communications, but I am unable to recommend publication of this version due to a number of concerns. I would be willing to review a significantly revised version of the manuscript, should the Editor allow a second round.

My first major concern is the wider message of the manuscript that should appeal to wider audiences. This is unclear in the manuscript: white-light flare ribbons, even circular ones, are not unprecedented. The totally original element of the manuscript is the gradual white-light flare ribbon, but what does it mean? The only evidence presented is the circular QSL footprint morphology but evidence of localized magnetic reconnection or evidence of downward-propagating Alfvén waves is lacking. If the authors cannot demonstrably adhere to the one or the other interpretation, even reporting the original effect and stating that its origin is unclear and should be investigated further in certain ways (a brief reference to these ways would be useful) would be a legitimate message.

Second, circular footprints in QSLs are reported relatively frequently for eruptive solar active regions. What is the distinguishing feature of this QSL that gives rise to the gradual white-light component is also unclear and is not discussed in the manuscript. In essence, what is the physics that makes this flare so rare, to the authors' assessment?

A number of more specific, but still significant, concerns follow below:

- It is hard to directly cross-check between Figures 1, 2, and 5 because the field of view is different in all three Figures. Making visual comparisons easier in some meaningful way would be very important.
- Figures 1 and 2 refer to the pre-flare situation, up to the flare peak. Showing some frames from the post-flare situation would also be important and useful as it would allow for a comparison between white-light emission before and after the event. In addition, frames g – i (insets, most notably) of Figure 1 are not explained.
- The terms "flux rope" and "sheared arcade" are often used interchangeably in the manuscript, without much consideration. However, they represent quite distinct features in the literature. The authors should decide what they believe or think exists above the sheared photospheric magnetic polarity inversion lines they discuss: a flux rope or a sheared arcade?
- The NLFF field extrapolation method used is not discussed at all in the manuscript. Which photospheric SDO/HMI vector magnetogram data have been used and what preprocessing, if any,

has been applied to them? We know that extrapolating for the coronal magnetic field is an ill-posed problem and different methods are known to give different results on identical photospheric boundary conditions. How certain are the authors that the OSLs presented in Figure 5 are not partially or fully due to the extrapolation method used? One should admit that the pre-flare situation in Figure 2 (rows c and d) arguably provides some support of this QSL scenario but the analysis needs to elaborate on this a bit more.

Some minor comments on the manuscript follow below:

- Section 1, p.1: "Although the physical parameters of stellar white-light flares are diverse...". What are these parameters, briefly, and in what sense are they "diverse"? An explanatory statement would be useful here.
- Section 2, p.2: "The spatial resolution [of RHESSI detectors 1,3,5,6,7,9] is about 3" ". Is this really the case, or is there a pixel size of 3" giving rise to a spatial resolution of 6"?"
- One repeatedly finds in the manuscript the noun "impulse" used as an adjective. I would recommend replacing it by "impulsive".
- Please make sure that the manuscript is scanned for typos because several seem to exist.

Dear Reviewers,

First we express our great thanks for all the comments, which have greatly improved our manuscript. As follows, we reply to all the questions and list all the additional changes in the revised version. Since the manuscript has been revised greatly and almost rewritten, we didn't use the bold font for the revisions.

Reviewer #1 (Remarks to the Author):

Solar flares release tremendous amount of energy and are often associated with other important solar events, such as sunquakes, filament eruptions, and coronal mass ejections (CMEs). The white-light (WL) flares represent the most energetic component of flare emission. Their properties in lower solar atmosphere are not fully understood in terms of the intriguing particle acceleration and radiative transfer processes. This paper presents a WL flare observation with a circular ribbon, which is very interesting and was not reported before. Its research will extend the frontier of existing knowledge on the energy release process of flares and its impact on the deep solar atmosphere. But the paper was prepared a bit roughly and needs polishing before publishing in Nature Communications.

(1) *Hinode* had a good coverage of this event, in high resolution, with a good pre-flare vector data at 00 UT. *Hinode* observations may miss some part of the ribbons, but should be compared, especially in G-band.

A: We checked the *Hinode* data carefully. Unfortunately, the data only cover the impulsive source regions. As shown in Reply-Figure 1, the field of view of the *Hinode* data covers nearly half of the southern region of the field of view of Figures 1, 2 and 5 in the manuscript. The time cadence is one minute in *Hinode* Ca II H images, which is similar to the ONSET H α waveband. Moreover, both data show a quite similar observational structure. The G-band was set at a low cadence in the operation plan with a cadence of only ten minutes, which does not capture the white-light flare. Therefore we do not include the *Hinode* data.

(2) I am not convinced for the lack of HXR for the second WL peak as shown in Figure 3. The *RHESSI* HXR 12-25 keV plot has a second peak, and the 25-100 keV seems to be somehow strange. Is the attenuation of *RHESSI* considered? It is hard to believe that WL emission without HXR signature since HXR is very sensitive, even to the micro-flares.

A: The second peak of *RHESSI* HXR 12-25 keV is due to a change of the various attenuators. We replotted the corrected *RHESSI* HXR flux in the revised Figure 3. The 25-100 keV was the sum of two passbands 25-50 and 50-100 keV in previous version. We replotted the corrected separate ones in the revised Figure 3. We adopted Clean, Pixon, Back Projection, and MEM NJIT algorithms to reconstruct the HXR sources, and the results show that the gradual white-light sources have no any obvious corresponding HXR sources. We also reconstructed the HXR sources and checked the flux in different white-light emission regions. The HXR flux in the gradual white-light source regions is within the noise level and is not reliable. Besides, we discussed with Drs. Wei Liu and Richard A. Schwartz for the accuracy and spatial resolution of the reconstructed HXR images. Our results are verified to be convincing.

(3) The authors identified the WL ribbon in the wavelengths of 360 nm and 425 nm. It is unable for me to evaluate the real contamination at these two specific observing channels due to lack of information on bandpass of the used filters.

A: The white-light vacuum tube has an aperture of 20 cm, working at the 3600 Å and 4250 Å continua with a band width of 15 Å. These wavebands are not contaminated by any absorption lines. We also added the description of these two bands in lines 74-75 in Sect. 2.1.

(4) It is better to add a figure of *Hinode*/SP magnetogram marking the kernels.

A: *Hinode*/SP magnetogram has a relatively high resolution, but as mentioned in our reply to Question (1), the *Hinode* field of view is not fully covering the whole flare site. Therefore, the *Hinode* magnetogram is not sufficient for

Reply-Figure 1. The red box shows the field of view of Figure 1 and Figure 2, respectively. The black box shows the field of view of the Hinode data.

magnetic field extrapolation. Besides, *Hinode*/SP magnetogram should be reconstructed from the Stokes parameters I, Q, U and V, which is much complex and dependent on the variation of reconstruction methods. We think that the *SDO*/HMI data are good enough for the analysis of the magnetic structure of the flaring region because the extrapolated 3D magnetic field does reveal some small magnetic flux ropes which may correspond to the white-light sources.

(5) With the derived contrast, some quantitative analysis could be done, such as temperature.

A: This is a good suggestion. However, the present data (two continuum windows at 3600 Å and 4250 Å) are not sufficient for derivation of reliable temperature increase. If we assume a blackbody emission and set the temperature $T=6000$ K in the photosphere, the maximum contrast corresponds to a temperature increase ΔT of 299 K for 3600 Å and 265 K for 4250 Å. Because the blackbody assumption is not accurate, we do not discuss this in the manuscript. In future work, we plan to make detailed radiative hydrodynamic simulations of the flare atmosphere to check more clearly what could happen in the lower atmosphere.

(6) In order to form the dome-spine structure, the circular ribbon should enclose a compact opposite magnetic polarity core emission. The paper did not discuss this part at all.

A: It is true that in a typical dome-spine structure, there should be a compact opposite magnetic polarity inside the circular ribbon. However, our event is not a typical dome-spine structure. There is neither a null point nor an inner spine since the polarity inversion line possesses an irregular path. So there is no core emission corresponding to the inner spine that is enclosed by the circular ribbon. We added the description of the 3D magnetic configuration in lines 177-181 in Sect. 2.3.

(7) The authors state that the cadence of ONSET data is one minute, how can they plot sub-minute data points in the light curves of Figure 3?

A: Actually, the cadence of ONSET is 15 - 60 s depending on the observation modes. Since the seeing is not good enough sometimes so that some blurred frames may appear. We deleted the blurred frames manually. We added the description on this issue in lines 78-79 in Sect. 2.1.

Reviewer #2 (Remarks to the Author):

The main claim of this paper is “This is the first report of this type of gradual white-light emission, which is not correlated with the HXR emission.” However, there is no discussion of the so-called Type II white-light flares that are not associated with HXR emission and appear in the late phase (e.g., Ding, M. D., Fang, C., & Yun, H. S. 1999, ApJ, 512, 454). Therefore, I cannot recommend publication in Nature Communications.

A: Thanks for pointing out this important question. Actually, the gradual white-light sources in our event and type II white-light flares are not equivalent concepts though they have some similar features. Type II white-light flares have two main characteristics (Fang & Ding 1995). One is that there is no obvious temporal correspondence between the continuum maximum and the HXR peak; the former could precede or lag the latter by several minutes. The other is that the Balmer jump is very weak or disappears. For the gradual white-light sources in our case, on the one hand they do have these two features; on the other hand, they show some new features that previous type II white-light flares do not have: a gentle rise phase (in some cases) and a gentle decay phase with a long lifetime. Such a gradual evolution of the white-light flare is a new finding that has not been reported in previous literatures. We added detailed description on this issue in lines 228-247 in Sect. 3.2.

Furthermore, a statistical analysis of the 4250 and 3600 enhancements is completely absent from the paper, and it is not convincing from Figure 3 that the black, blue, and yellow light curves in 4250 experience a statistically significant gradual increase (e.g., how do the light curves of other regions of plage look compared to these light curves?).

A: We selected a quiet region with a size of 150×150 pixels near the flare site and then calculated the mean standard deviation of the intensity from 03:12 to 03:42 UT in order to estimate the observation error in the continua. The mean standard deviation is $2.44 \text{ count s}^{-1} \text{ pixel}^{-1}$, representing an observation error of about 2.5% at 3600 Å and 4250 Å. We added the description on the statistical analysis of two wavebands in lines 79-82 in Sect. 2.1.

Since the light curves in the previous manuscript are plotted within a short time period, it is hard to distinguish the gradual and the impulsive sources clearly. We drew a new version of the Figure 3 with a much longer period from 03:13 to 03:35 UT, which can show the complete evolution from the pre-flare to the post-flare phases. Then we can see an increase in the three gradual sources which is over the observation error. As the reviewer suggested, we also added the light curves of a plage for a comparison (gray lines with plus signs in Figure 3(c) and (d)).

In the black, blue, and yellow curves in the 3600 light curve, only the black appears to increase, and this increase occurs at the same time as the red and orange increase (thus all would be impulsive).

A: To make a reliable result, we reanalyse the data and replot the light curves of the five patches in the new version of Figure 3. We also plot the maximum continuum contrasts at the five white-light patches at 3600 Å and 4250 Å in Figure 4 in the new version. Besides, we measured quantitatively the lifetime and the rise time of the white-light emissions at different patches which are listed in Table 1. The results show that the gradual sources (pointed by the blue, navy and black arrows in Figure 1) have a much longer lifetime than the impulsive sources (pointed by the red and orange arrows in Figure 1). It is true that the black curve increases at the same time as the red and orange curves; but the former has a relatively long decay phase that lasts longer than four minutes. Therefore, we think the sources indicated by black, navy and blue arrows are basically gradual sources which have an origin different from the impulsive sources (red and orange). We added this description in detail in lines 131-145 in Sect. 2.2. We also discuss the slight difference between the three gradual sources in lines 238-253 in Sect. 3.2.

There are also many statements about the sources of energy deposition (in general and specifically to this flare) that are qualitative, without appropriate references, and unjustified.

A: We have reprocessed the observation data and added some quantitative calculation, such as the lifetime of the white-light sources (in lines 137-145 in Sect. 2.2) and the check of the existence of a Balmer jump (in lines 146-153 in Sect. 2.2). We discussed the new results and compared these with previous works in order to give the possible interpretation for the mechanisms that generate these white-light sources. In particular, we compared the five white-light

sources with the magnetic structure and found that the impulsive and gradual sources are located at different positions and formed through different types of reconnection: the impulsive sources originate from reconnection caused by the eruption of a flux rope, while the gradual sources may originate from QSL reconnection. We add a detailed discussion and references on this issue in lines 254-281 in Sect. 3.2.

Reviewer #3 (Remarks to the Author):

The manuscript reports on an apparently rare solar flare that shows an observable signature in white light. While white-light flares statistically belong to the largest flares (GOES M- and X-classes), and hence represent a slim minority of the solar flare population, the rarity of the flare studied here is a roughly circular white-light ribbon consisting of two parts: an impulsive one, correlating with the flare's X-ray emission, and a gradual one, with no apparent correlation to X-rays. The gradual white-light flare emission, contributing much of the circular ribbon, is interpreted as either (slow) magnetic reconnection or downward-propagating Alfvén waves originating in the corona, with both mechanisms leading to emission along the circular footprint of a magnetic quasi-separatrix layer (QSL) bounding the entire flaring magnetic configuration.

The manuscript's topic is quite interesting, while analysis combines science-grade data from an array of ground- and space-based instruments: Near-Infrared Solar Eruption Tracer (ONSET), RHESSI and two telescopes (HMI and AIA) onboard the Solar Dynamics Observatory (SDO) mission. At the same time, it uses a nonlinear force-free (NLFF) field extrapolation for the analysis of QSLs. Its originality could potentially make it worthy of publication at Nature Communications, but I am unable to recommend publication of this version due to a number of concerns. I would be willing to review a significantly revised version of the manuscript, should the Editor allow a second round.

My first major concern is the wider message of the manuscript that should appeal to wider audiences. This is unclear in the manuscript: white-light flare ribbons, even circular ones, are not unprecedented. The totally original element of the manuscript is the gradual white-light flare ribbon, but what does it mean? The only evidence presented is the circular QSL footprint morphology but evidence of localized magnetic reconnection or evidence of downward-propagating Alfvén waves is lacking. If the authors cannot demonstrably adhere to the one or the other interpretation, even reporting the original effect and stating that its origin is unclear and should be investigated further in certain ways (a brief reference to these ways would be useful) would be a legitimate message.

A: Thanks for the valuable suggestions. As is well known, in previous white-light studies, the white-light emissions are always found in a compact region (Metcalf et al. 2003; Xu et al. 2006; Fletcher et al. 2007; Hao et al. 2012; Penn et al. 2016). In our case, however, the white-light emissions form ribbons that constitute a circular configuration. In fact, the white-light emissions are generated in several separate regions of a circular configuration with very different spectral properties, *i.e.*, with and without the Balmer jump. This indicates that the white-light emissions are not caused by the QSL reconnection in the large scale spine-fan field lines which generate the typical circular flares as proposed by previous studies (Masson et al. 2009; Reid et al. 2012; Wang & Liu 2012; Sun et al. 2013; Jiang et al. 2013). The possible scenario is that these white-light emissions are generated by reconnection at different heights with different energy transport mechanisms so that they display quite different evolution styles. More interesting is that we detect two kinds of white-light patches, impulsive ones and gradual ones. While the impulsive ones are like most previous cases, the gradual ones represent a new finding. In particular, the gradual sources are not identical to the classical type II white-light flares in that the former have a gradual evolution and a long lifetime. We discuss the different origins of impulsive sources and gradual sources in lines 254-281 in Sect. 3.2. The quasi-circular white-light emission, different kinds of white-light sources, and the specific magnetic structure make this event a unique one so far (we mention this in lines 64-66 in Sect. 1).

Although we propose some possible origins of the white-light sources, it is difficult to verify them with the present observations, as pointed out by the reviewer. For example, we expect future coordinated observations of white-light

flares, in particular including high-cadence spectral observations of chromospheric lines like $H\alpha$, to test the existence of Alfvén waves (Jess et al. 2009), which we suppose is a possible mechanism for the gradual sources. Besides the observations, radiative hydrodynamic simulations are also required to elucidate the different roles of energy transport mechanisms including electron beam bombardment, heat conduction, Alfvén waves, and the radiative backwarming. Such calculations are beyond the scope of this paper and will be done in future work.

Second, circular footprints in QSLs are reported relatively frequently for eruptive solar active regions. What is the distinguishing feature of this QSL that gives rise to the gradual white-light component is also unclear and is not discussed in the manuscript. In essence, what is the physics that makes this flare so rare, to the authors assessment?

A: Yes, there are quite some circular ribbon flares reported so far. In usual cases, the dome-shaped fan portrays a closed separatrix surface, and the inner and outer spine field lines in different connectivity domains meet at the null point. When perturbations give rise to the null point reconnection, flare emissions at the footpoints of the fan field lines would constitute a closed or open circle-like flare ribbon. However, our case is somehow different from this scenario. In our case, the QSL reconnection may happen around the twisted field lines, not only in the dome structure. We speculate the consecutive causes for the impulsive and gradual white-light emissions as follows. The QSL reconnection may take place in several separate peripheral twisted field lines producing separate gradual white-light emissions in the beginning (the three sources pointed by the blue, navy and black arrows in Figure 1(a)). These reconnections together with the reconnection in the dome probably facilitate the eruption of the core twisted field lines, which leads to two impulsive white-light sources at the conjugate footpoints of a flux rope (the two sources pointed by the orange and red arrows in Figure 1(b) and (c)). The eruption then disturbs the above dome structure and the lower layer magnetic field configuration, further sustaining the QSL reconnection for a long time to generate the continuous gradual white-light emissions. Note that, in the early phase of the QSL reconnection, it is possible that some energetic electrons are also generated in the lower atmosphere, which are thermalized locally during a very short time because of the high density there. The local thermalization and the resulting heating may explain the relatively rapid increase in white-light among the two of the three gradual sources, *i.e.*, the two sources marked by the navy and black arrows in Figure 1(b) and (c). We think the rareness of this flare is that its specific magnetic structure leads to sequential reconnections that produce different kinds of white-light sources. We include the above discussions in lines 254-268 in Sect. 3.2.

A number of more specific, but still significant, concerns follow below:

- **It is hard to directly cross-check between Figures 1, 2, and 5 because the field of view is different in all three Figures. Making visual comparisons easier in some meaningful way would be very important.**

A: We replotted Figures 1, 2 and 5 in the revised manuscript. The fields of view of Figure 1 and Figure 2 are now the same. Due to the correction for projection effects and the transformation between the observation coordinate and the heliographic coordinate, the field of view of Figure 5 has some distortions compared with that of Figure 1. However, the field of view of new Figure 5 is roughly the same as that of Figure 1 and Figure 2.

- **Figures 1 and 2 refer to the pre-flare situation, up to the flare peak. Showing some frames from the post-flare situation would also be important and useful as it would allow for a comparison between white-light emission before and after the event. In addition, frames g - i (insets, most notably) of Figure 1 are not explained.**

A: We added the frames from the pre-flare to the post-flare situations in Figures 1 and 2 in the revised manuscript. Since many authors adopted the *SDO*/HMI 6173 Å intensity to search for white-light flare candidates, here we also add these images for comparison. The enlarged frames are the base-difference images, from which the white-light kernels can be seen more clearly. We also added the description of panels i to l (*i.e.*, frames g - i in previous version) in the caption of Figure 1.

- **The terms “flux rope” and “sheared arcade” are often used interchangeably in the manuscript, without much consideration. However, they represent quite distinct features in the literature. The**

Reply-Figure 2. The Q distribution on the solar surface by NLFFF extrapolation (left panel) and potential extrapolation (right panel).

authors should decide what they believe or think exists above the sheared photospheric magnetic polarity inversion lines they discuss: a flux rope or a sheared arcade?

A: By carefully checking the magnetic structure, we think “flux rope” is more appropriate in describing the situation. Therefore, “flux rope” is used throughout the revised manuscript.

- The NLFF field extrapolation method used is not discussed at all in the manuscript. Which photospheric SDO/HMI vector magnetogram data have been used and what preprocessing, if any, has been applied to them? We know that extrapolating for the coronal magnetic field is an ill-posed problem and different methods are known to give different results on identical photospheric boundary conditions. How certain are the authors that the QSLs presented in Figure 5 are not partially or fully due to the extrapolation method used? One should admit that the pre-flare situation in Figure 2 (rows c and d) arguably provides some support of this QSL scenario but the analysis needs to elaborate on this a bit more.

A: We added the description of the NLFFF extrapolation method and the level of acceptance of the results in lines 155-168 in Sect. 2.3. The calculated Q distribution on the solar surface is supported by the polarity inversion lines in Figure 1 and the flare evolution in the panels of Figure 2 in the revised manuscript. The flare emissions at the footpoints of the fan field lines would constitute a closed or open circle-like flare ribbon. The evidence of a fan-spine configuration has been reported by several authors (Masson et al. 2009; Reid et al. 2012; Wang & Liu 2012; Deng et al. 2013; Sun et al. 2013; Jiang et al. 2013, 2014; Liu et al. 2015; Yang et al. 2015; Joshi et al. 2015). Besides, we also calculated the Q distribution on the solar surface by the potential extrapolation, as shown in the right panel in Reply-Figure 2. It shows quite similar outlines to that of the NLFFF extrapolation. These mean that our results are reliable.

Some minor comments on the manuscript follow below:

- Section 1, p.1: “Although the physical parameters of stellar white-light flares are diverse”. What

are these parameters, briefly, and in what sense are they “diverse”? An explanatory statement would be useful here.

A: The stellar white-light flares are very powerful with a total energy of $10^{33} - 10^{39}$ erg, $10 - 10^7$ times (Schaefer et al. 2000; Davenport 2016) larger than the energy of the largest solar flares, 10^{32} erg (Shibata & Yokoyama 2002), and their duration is from several minutes to hours (Schaefer et al. 2000; Shibayama et al. 2013). Magnetic field with a few kG covers a larger area on the stellar surface than that of the solar counterpart (Gershberg 2005; He et al. 2015). Since our main focus is the solar white-light flare, we briefly discussed the comparison between the solar and stellar white-light flares in lines 41-44 in Sect.1.

• **Section 2, p.2:** “The spatial resolution [of RHESSI detectors 1,3,5,6,7,9] is about $3''$ ”. Is this really the case, or is there a pixel size of $3''$ giving rise to a spatial resolution of $6''$?

A: The spatial resolution is $3''$, not the pixel size. The different detectors have different spatial resolutions (FWHMs). The detectors 1, 3, 5, 6, 7, and 9 are used in the image reconstruction in our case and the approximate effective resolution of such a combination of RHESSI detectors is about $3''$.

• **One repeatedly finds in the manuscript the noun “impulse” used as an adjective. I would recommend replacing it by “impulsive”.**

A: We replaced the “impulse” by “impulsive”.

• **Please make sure that the manuscript is scanned for typos because several seem to exist.**

A: We checked the manuscript for typos and corrected them.

Thank you very much again!

Best regards,

Qi Hao and coauthors

REFERENCES

- Davenport, J. R. A. 2016, ApJ, 829, 23
 Deng, N., Tritschler, A., Jing, J., et al. 2013, ApJ, 769, 112
 Fang, C., & Ding, M. D. 1995, A&AS, 110, 99
 Fletcher, L., Hannah, I. G., Hudson, H. S., & Metcalf, T. R. 2007, ApJ, 656, 1187
 Gershberg, R. E. 2005, Solar-Type Activity in Main-Sequence Stars, doi:10.1007/3-540-28243-2
 Hao, Q., Guo, Y., Dai, Y., et al. 2012, A&A, 544, L17
 He, H., Wang, H., & Yun, D. 2015, ApJS, 221, 18
 Jess, D. B., Mathioudakis, M., Erdélyi, R., et al. 2009, Science, 323, 1582
 Jiang, C., Feng, X., Wu, S. T., & Hu, Q. 2013, ApJL, 771, L30
 Jiang, C., Wu, S. T., Feng, X., & Hu, Q. 2014, ApJ, 780, 55
 Joshi, N. C., Liu, C., Sun, X., et al. 2015, ApJ, 812, 50
 Liu, C., Deng, N., Liu, R., et al. 2015, ApJL, 812, L19
 Masson, S., Pariat, E., Aulanier, G., & Schrijver, C. J. 2009, ApJ, 700, 559
 Metcalf, T. R., Alexander, D., Hudson, H. S., & Longcope, D. W. 2003, ApJ, 595, 483
 Penn, M., Krucker, S., Hudson, H., et al. 2016, ApJL, 819, L30
 Reid, H. A. S., Vilmer, N., Aulanier, G., & Pariat, E. 2012, A&A, 547, A52
 Schaefer, B. E., King, J. R., & Deliyannis, C. P. 2000, ApJ, 529, 1026
 Shibata, K., & Yokoyama, T. 2002, ApJ, 577, 422
 Shibayama, T., Maehara, H., Notsu, S., et al. 2013, ApJS, 209, 5
 Sun, X., Hoeksema, J. T., Liu, Y., et al. 2013, ApJ, 778, 139
 Wang, H., & Liu, C. 2012, ApJ, 760, 101
 Xu, Y., Cao, W., Liu, C., et al. 2006, ApJ, 641, 1210
 Yang, K., Guo, Y., & Ding, M. D. 2015, ApJ, 806, 171

Reviewers' comments:

Reviewer #1 (Remarks to the Author):

The 2nd version of manuscript has been improved significantly. All my questions and concerns have been addressed. This research will extend the frontier of existing knowledge on the energy release process of flares and its impact on the deep solar atmosphere. I recommend this paper to be published in Nature Communications.

Reviewer #2 (Remarks to the Author):

The authors have satisfactorily addressed the comments and concerns. However, there are three major outstanding issues with the author's reply and additions to the manuscript:

1) The authors emphasize that the gradual source that coincides with the soft X-ray maximum is a new aspect not seen in type II flares before: "Such a gradual evolution of the white-light are is a new nding that has not been reported in previous literatures." On the contrary, gradual white-light flares have been observed: see Matthews et al. 2003 A&A 409, 1107. The manuscript should include a discussion of the results of this work.

2) Due to the timing of the soft X-ray peak with the gradual source, one wonders why the authors discuss the role of other mechanisms, such as Alfvén wave energy transport, that have no support from the presented observations. The only logical conclusion from the data is that there may be a role in the soft X-ray backwarming and the heating in the gradual source.

3) The manuscript claims there is a Balmer jump in Figure 4, but it is not possible to determine this with a course energy distribution. Do the data rule out a blackbody curve fit to the kernels indicated by the red and orange symbols? Please clarify.

Reviewer #3 (Remarks to the Author):

Referee's Second Report

NCOMMS-16-29810: A Circular White-Light Flare with Impulsive and Gradual White-Light Sources, by Hao, Q., Yang, K., Cheng, X., Guo, Y., Fang, C., Ding, M. D., Chen, P. F. and Li, Z.

I thank the authors for their responses and revisions to initial comments, that have significantly improved the manuscript. While I now have a much better understanding of the observations and analysis, I continue to feel that the case for this manuscript to appear in Nature Communications is rather underwhelming. This is not because of trivialities in the context presented but because the manuscript does not seem to highlight well enough its unique element and the broader picture this element suggests: the simultaneous existence of impulsive and gradual white-light kernels in a nearly circular-ribbon white-light flare. The larger picture is apparently that the generally accepted thick-target bremsstrahlung, that would be responsible for the impulsive kernels, is not sufficient to interpret this particular event and similar events. Downward-propagating Alfvén waves (or other mechanism[s], possibly) acting jointly with energetic electrons bombarding the lower solar atmosphere is a plausible interpretation but, in fact, the answer is unknown, because observations are not sufficient to resolve this. The case will remain unsolved until future observations decipher it.

A clear narrative that would immediately convey this picture, or the picture that the authors wish to promote as their bottom-line conclusion, should appear right from the abstract.

Following the above top-level suggestion, I still have some comments / suggestions / questions on the text and figures. Therefore, I would be willing to review another manuscript revision addressing them. My comments are the following:

* General: the bright white-light areas seem to be described by means of three different terms in the text: "patches", "sources" and "kernels". Each of these terms appears numerous times in different parts of the manuscript. For consistency, the authors need to decide which term will be used primarily, and why.

* I.87: [Figures 1b, 1c]: "These bright patches form a circular morphology". From Figures 1b, 1c, it appears that the morphology of the bright patches is more semi-circular than circular, as the northernmost part of the configuration does not seem to generate any such patches.

* Section 2.2, Figure 3 (top) and related discussion: the GOES SXR component differs from the RHESSI HXR ones not only in the peak time, but also in the onset time: the SXR onset time roughly coincides with the HXR peak time. This feature is striking in Figure 3 but is not discussed at all in the manuscript. Is it incidental or not? A possible interpretation and relevant citations, if applicable, would be useful.

* The white-light sources indicated by the black arrows in Figures 1a-d seem to follow a different type of behavior (Table 1), namely one that would classify them as between "impulsive" and "gradual" sources. In fact, inspecting their mean lifetimes and rise times, they are closer to "impulsive" than to "gradual" sources. Instead, they are characterized as "gradual". I would question this and prompt the authors to reassess this interpretation.

* More with respect to Table 1, the characteristic onset time of each feature, including the GOES SXR component, would also be revealing.

* The Discussion, Section 3, includes two statements that generate and promote vagueness:
- I.201 [re: impulsive component]: "We surmise that the reconnection probably occurs in the lower corona."

- II.269-270 [re: gradual component]: "The above discussions suggest a relatively lower site of magnetic reconnection ..."

The manuscript needs to explain what is meant by "lower corona" and what by a "relatively lower site" as the seat of magnetic reconnection. A scale height with respect to some characteristic scale height, or an absolute altitude range in Mm from $\tau=0$ for both cases would clarify this somewhat. Moreover, it is unclear whether the two reconnection sites mentioned are the same. Would different reconnection sites mean that impulsive and gradual components have a different origin within the same flare? Or is it a single major reconnection event that, depending on position, would give rise to both components? I feel that the manuscript does not do much to clarify these questions or, at least, to explicitly state that they are open questions.

* A (minor) comment with respect to Figure 5: its field-of-view (65" x 65") continues to not be identical to the respective FOVs in Figures 1 and 2 (75" x 75"). In case the authors simply missed to give identical FOVs to Figures 1, 2 and 5 as per one of my initial comments, they could update Figure 5 in the revision. Otherwise, this is not too important now as one is able to cross-check between figures in the revised manuscript.

Dear Reviewers,

Thank you very much for the valuable comments. We have revised the paper accordingly and tried to clarify each point. We list our replies to the comments as follows. In the revised version, all the changes appear in blue-boldface.

Reviewer #1 (Remarks to the Author):

The 2nd version of manuscript has been improved significantly. All my questions and concerns have been addressed. This research will extend the frontier of existing knowledge on the energy release process of flares and its impact on the deep solar atmosphere. I recommend this paper to be published in Nature Communications.

A: Thank you for your recommendation.

Reviewer #2 (Remarks to the Author):

The authors have satisfactorily addressed the comments and concerns. However, there are three major outstanding issues with the author’s reply and additions to the manuscript:

1) The authors emphasize that the gradual source that coincides with the soft X-ray maximum is a new aspect not seen in type II flares before: “Such a gradual evolution of the white-light are is a new finding that has not been reported in previous literatures.” On the contrary, gradual white-light flares have been observed: see Matthews et al. 2003 A&A 409, 1107. The manuscript should include a discussion of the results of this work.

A: Thank you for reminding us of this relevant paper on the gradual white-light source. We have added a discussion on this point in lines 243–248 in Section 3.2.

2) Due to the timing of the soft X-ray peak with the gradual source, one wonders why the authors discuss the role of other mechanisms, such as Alfvén wave energy transport, that have no support from the presented observations. The only logical conclusion from the data is that there may be a role in the soft X-ray backwarming and the heating in the gradual source.

A: We agree with the point and have added the soft X-ray backwarming as one of the possible mechanisms for the interpretation of the gradual kernels in lines 295–297 in Section 3.2, together with the possible role by Alfvén waves. A trivial reason for the latter is that the gradual sources are found to appear at the QSL layers around some flux ropes, where probably Alfvén waves can be excited. We admit that these mechanisms are speculative which the present data cannot confirm. We expect more observations in the future and numerical simulations to test them.

3) The manuscript claims there is a Balmer jump in Figure 4, but it is not possible to determine this with a course energy distribution. Do the data rule out a blackbody curve fit to the kernels indicated by the red and orange symbols? Please clarify.

A: Yes, this is a good point. We propose that without detailed continuum spectra, whether there is a Balmer jump can be judged by comparing the observed intensity ratio to that predicted by the blackbody emission. To clarify this, we have made a test by assuming an isothermal blackbody source to account for the emissions at the two wavebands. We found that a blackbody source can produce a ratio of intensity enhancement $(\Delta I_{3600}/I_{3600})/(\Delta I_{4250}/I_{4250})$ being about 1.17 when $T = 6000$ K. However, the observed value of this ratio is about 1.45 for the kernel indicated by the orange symbol and about 1.20 for the kernel indicated by the red symbol. Thus, an isothermal blackbody emission cannot fit the intensities at the two wavebands simultaneously. This implies that the emissions at the two wavebands are from different heights with different temperatures or the emission layer deviates from local thermodynamic equilibrium. These conditions may result in an intensity jump at 3600 Å relative to that at 4250 Å which is greater than what is predicted by the blackbody emission. We added this description in lines 162–165 in Section 2.2.

Reply-Figure 1. (a) The *GOES* SXR 1–8 Å flux (orange) and its time derivative (black) for the white-light flare. (b) The *RHESSI* HXR 12–25 keV (black), 25–50 keV (blue), 50–100 keV (red), and 100–300 keV (yellow) for the white-light flare. The long dashed vertical lines show the onset time of the SXR flux, the solid vertical lines show the peak time of the HXR flux and the short dashed vertical lines show the peak time of the SXR flux time derivative.

Reviewer #3 (Remarks to the Author):

I thank the authors for their responses and revisions to initial comments, that have significantly improved the manuscript. While I now have a much better understanding of the observations and analysis, I continue to feel that the case for this manuscript to appear in *Nature Communications* is rather underwhelming. This is not because of trivialities in the context presented but because the manuscript does not seem to highlight well enough its unique element and the broader picture this element suggests: the simultaneous existence of impulsive and gradual white-light kernels in a nearly circular-ribbon white-light flare. The larger picture is apparently that the generally accepted thick-target bremsstrahlung, that would be responsible for the impulsive kernels, is not sufficient to interpret this particular event and similar events. Downward-propagating Alfvén waves (or other mechanism[s], possibly) acting jointly with energetic electrons bombarding the lower solar atmosphere is a plausible interpretation but, in fact, the answer is unknown, because observations are not sufficient to resolve this. The case will remain unsolved until future observations decipher it.

A clear narrative that would immediately convey this picture, or the picture that the authors wish to promote as their bottom-line conclusion, should appear right from the abstract.

A: Thanks for the valuable suggestions. We fully agree and have rewritten the abstract with a clear narrative of this event following these suggestions (please see the latter part of the abstract).

Following the above top-level suggestion, I still have some comments / suggestions / questions on the text and figures. Therefore, I would be willing to review another manuscript revision addressing them. My comments are the following:

* **General:** the bright white-light areas seem to be described by means of three different terms in the

text: “patches”, “sources” and “kernels”. Each of these terms appears numerous times in different parts of the manuscript. For consistency, the authors need to decide which term will be used primarily, and why.

A: In fact, these terms refer to the same thing. For consistency, we have unified the terms to “kernels” in the whole text, since this term has been more frequently used in literatures.

* 1.87: [Figures 1b, 1c]: “These bright patches form a circular morphology”. From Figures 1b, 1c, it appears that the morphology of the bright patches is more semi-circular than circular, as the northernmost part of the configuration does not seem to generate any such patches.

A: We changed this description into “These bright kernels form a semi-circular morphology.” in line 90 in Section 2.1.

* Section 2.2, Figure 3 (top) and related discussion: the *GOES* SXR component differs from the *RHESSI* HXR ones not only in the peak time, but also in the onset time: the SXR onset time roughly coincides with the HXR peak time. This feature is striking in Figure 3 but is not discussed at all in the manuscript. Is it incidental or not? A possible interpretation and relevant citations, if applicable, would be useful.

A: We have made a careful measurement of the SXR and HXR light curves. The *GOES* SXR onset time is about 03:19:39 UT (defined as the time from which the flux increases monotonically) and the peak time is about 03:23:54 UT. We also calculated the *GOES* SXR flux time derivative, as shown in Reply-Figure 1. The peak time of the SXR time derivative is 03:21:30 UT. The HXR peak times at the four wavebands are 03:21:32 UT (12–25 keV), 03:21:08 UT (25–50 keV), 03:21:04 UT (50–100 keV) and 03:21:04 (100–300 keV), respectively. So the HXR peak times lag the SXR onset time more than one minute. However, the maximum SXR time derivative is roughly coincident with the HXR peak (especially for that of 12–25 keV), implying that the Neupert effect well applies for this event (Hudson 1991; Dennis & Zarro 1993; Veronig et al. 2005). We added a short description in lines 120–122 in Section 2.2.

* The white-light sources indicated by the black arrows in Figures 1a-d seem to follow a different type of behavior (Table 1), namely one that would classify them as between “impulsive” and “gradual” sources. In fact, inspecting their mean lifetimes and rise times, they are closer to “impulsive” than to “gradual” sources. Instead, they are characterized as “gradual”. I would question this and prompt the authors to reassess this interpretation.

A: Thank you for reminding us of this character. For the kernel indicated by the black arrow, its rising time is similar to that of the impulsive kernels and the decay phase is quite long like that of the gradual kernels. We thus classify the black one as an “intermediate” kernel and have revised the relevant description in lines 140–142 and 152–153 in Section 2.2 and made a short discussion in lines 270–274 and 286–291 in Section 3.2.

* More with respect to Table 1, the characteristic onset time of each feature, including the *GOES* SXR component, would also be revealing.

A: We added the onset time of each feature and the *GOES* SXR emission in Table 1 and a description in lines 153–159 in Section 2.2.

* The Discussion, Section 3, includes two statements that generate and promote vagueness:

- 1.201 [re: impulsive component]: “We surmise that the reconnection probably occurs in the lower corona.”

- 11.269–270 [re: gradual component]: “The above discussions suggest a relatively lower site of magnetic reconnection ...”

The manuscript needs to explain what is meant by “lower corona” and what by a “relatively lower site” as the seat of magnetic reconnection. A scale height with respect to some characteristic scale height, or an absolute altitude range in Mm from $\tau=0$ for both cases would clarify this somewhat. Moreover, it is unclear whether the two reconnection sites mentioned are the same. Would different

reconnection sites mean that impulsive and gradual components have a different origin within the same flare? Or is it a single major reconnection event that, depending on position, would give rise to both components? I feel that the manuscript does not do much to clarify these questions or, at least, to explicitly state that they are open questions.

A: As the Reviewer pointed out, the present observations are not sufficient for determining where the reconnection sites are and how the released energy is transported to the white-light emission height. Therefore, we have deleted the relevant descriptions for the impulsive component in Section 3.1 and the gradual component in Section 3.2 regarding the reconnection heights. From the magnetic structure, we only find a slight difference in the magnetic environments that the impulsive kernels appear at the footpoints of a flux rope while the gradual kernels appear at the QSL layers around some flux ropes (5th paragraph in Sect. 3.2). Regarding the heating mechanisms, we agree that while it is plausible that the impulsive component is powered by the energy deposition of electron beam bombardment with the additional help of backwarming, the origins of the gradual sources remain open questions. We propose two possible mechanisms for the interpretation of the gradual component. We rephrased some sentences in lines 292–298 in Section 3.2.

*** A (minor) comment with respect to Figure 5: its field-of-view ($65'' \times 65''$) continues to not be identical to the respective FOVs in Figures 1 and 2 ($75'' \times 75''$). In case the authors simply missed to give identical FOVs to Figures 1, 2 and 5 as per one of my initial comments, they could update Figure 5 in the revision. Otherwise, this is not too important now as one is able to cross-check between figures in the revised manuscript.**

A: We replotted Figure 5 so that the field of view is now equivalent to that of Figures 1 and 2.

Other changes

- 1). We rewrote some sentences in lines 23–31 in Section 1 to describe the general properties of solar flares.
- 2). We rewrote some sentences in the last paragraph in lines 60–68 in Section 1 in order to highlight our results.
- 3). We deleted the Summary Section in the previous version and give a concise summary in lines 306–312 at the end of the Discussion Section.

Thank you very much again!

Best regards,

Qi Hao and coauthors

REFERENCES

- Dennis, B. R., & Zarro, D. M. 1993, SoPh, 146, 177
 Hudson, H. S. 1991, in BAAS, Vol. 23, Bulletin of the American
 Astronomical Society, 1064
 Veronig, A. M., Brown, J. C., Dennis, B. R., et al. 2005, ApJ,
 621, 482

Reviewers' comments:

Reviewer #2 (Remarks to the Author):

The authors have fully addressed by concerns #1 and #2, but only partially addressed my third point about fitting a blackbody to the data. The authors have only excluded a $T=6000$ K blackbody, but what about a $T=8000$ K blackbody or $T=10,000$ K blackbody? There is still no discussion about the necessity for better spectra to rule out the blackbody hypothesis. The authors should state explicitly which temperature(s) blackbody they have excluded. After this issue is addressed, I recommend publication.

Reviewer #3 (Remarks to the Author):

Referee's Third Report

NCOMMS-16-29810: A Circular White-Light Flare with Impulsive and Gradual White-Light Sources, by Hao, Q., Yang, K., Cheng, X., Guo, Y., Fang, C., Ding, M. D., Chen, P. F. and Li, Z.

Once again, I thank the authors for taking steps to improve their manuscript in response to my comments and concerns. These issues are now more or less sufficiently addressed in the manuscript. I am only left with a few minor comments, mostly pertinent to the newly added text. Provided that these comments are addressed - I do not require to see the manuscript again - I am happy to recommend it for publication in Nature Communications.

List of comments:

* The references to space weather (abstract; first par. in the Introduction) and coronal mass ejections (Introduction) are not necessary here - in fact, there can even be misleading, as this is the first and the last time that they are made in the manuscript. Interpreting white light kernels in flares does not show a (immediate, at least) benefit toward understanding CMEs or improving space weather forecasting.

* Introduction, ll. 67-68: "... unique event among white-light flares ... which provides a good example to explore the physical mechanisms for the white-light continuum emissions". If the studied flare is that rare, then probably it is the opposite, namely, the exception rather than the rule in explaining white-light emission. I would erase the second part of the sentence, from "which provides" onwards.

* I strongly encourage the authors to once again work on the language of the manuscript, particularly the contents of the newly conceived abstract, in order to make it more attractive and interesting to the reader.

Dear Reviewers,

Thank you very much for the valuable comments. We list our replies to the comments as follows. In the revised version, all the changes appear in blue-boldface.

Reviewer 2 (Remarks to the Author):

The authors have fully addressed by concerns 1 and 2, but only partially addressed my third point about fitting a blackbody to the data. The authors have only excluded a T=6000 K blackbody, but what about a T=8000 K blackbody or T=10,000 K blackbody? There is still no discussion about the necessity for better spectra to rule out the blackbody hypothesis. The authors should state explicitly which temperature(s) blackbody they have excluded. After this issue is addressed, I recommend publication.

A: Thanks for this comment. We have made more calculations. If the observed continuum spectra originate from a blackbody source, then theoretically we can calculate the ratio of intensity enhancement as $C_{3600}/C_{4250} = (\Delta I_{3600}/I_{3600})/(\Delta I_{4250}/I_{4250}) \approx (B'_{3600}/B_{3600})/(B'_{4250}/B_{4250})$, because the temperature increase is small (less than 300 K for explaining the observed continuum contrasts). We vary the temperature of the blackbody source with a reasonable range of 5000–10000 K (assuming the source in the upper photosphere or lower chromosphere). The calculated contrast ratio, together with the ratios observed for two white-light kernels, are shown in Reply-figure 1. It shows that the blackbody emission predicts a contrast ratio of no larger than 1.18. However, from observations, this ratio amounts to 1.45 and 1.20 for the two kernels indicated by the orange and red arrows, respectively. This implies that the white-light emissions at these kernels can unlikely be accounted for by an isothermal blackbody source, and that there exists a Balmer jump in the continuum spectra. We have added a discussion on this point in lines 168–177.

Reviewer 3 (Remarks to the Author):

Once again, I thank the authors for taking steps to improve their manuscript in response to my comments and concerns. These issues are now more or less sufficiently addressed in the manuscript. I am only left with a few minor comments, mostly pertinent to the newly added text. Provided that these comments are addressed - I do not require to see the manuscript again - I am happy to recommend it for publication in Nature Communications.

* The references to space weather (abstract; first par. in the Introduction) and coronal mass ejections (Introduction) are not necessary here - in fact, there can even be misleading, as this is the first and the last time that they are made in the manuscript. Interpreting white light kernels in flares does not show a (immediate, at least) benefit toward understanding CMEs or improving space weather forecasting.

A: Thanks for this suggestion. We have deleted the relevant content regarding space weather and coronal mass ejections in the abstract and main text.

* Introduction, ll. 67-68: "... unique event among white-light flares ... which provides a good example to explore the physical mechanisms for the white-light continuum emissions". If the studied flare is that rare, then probably it is the opposite, namely, the exception rather than the rule in explaining white-light emission. I would erase the second part of the sentence, from "which provides" onwards.

A: . We have deleted the second part of this sentence (lines 57–59).

Reply-Figure 1. The ratio of intensity enhancement C_{3600}/C_{4250} observed for the white-light kernels indicated by the orange and red arrows (the orange and red lines), compared with the value calculated from a blackbody source with a reasonable temperature range of 5000–10000 K (black line).

*** I strongly encourage the authors to once again work on the language of the manuscript, particularly the contents of the newly conceived abstract, in order to make it more attractive and interesting to the reader.**

A: We have made some revisions on the language and the wording of the manuscript. We hope the new version is more attractive and interesting to the reader.

Other changes

We have reorganized the subsections of the manuscript in order to comply with the editorial policies and the format requirements of the journal. Such editorial revisions and some language corrections are not highlighted in the manuscript since the scientific contents are not changed.

Thank you very much again!

Best regards,

Qi Hao and coauthors

REVIEWERS' COMMENTS:

Reviewer #2 (Remarks to the Author):

The paper now clearly states why the blackbody hypothesis is ruled out. I recommend publication of this article.

Dear Reviewers,

Thank you very much for the valuable comments and constructive suggestions.

Reviewer 2 (Remarks to the Author):

The paper now clearly states why the blackbody hypothesis is ruled out. I recommend publication of this article.

A: Thank you very much for your recommendation.

Thank you very much again!

Best regards,

Qi Hao and coauthors